# Fast-rate PAC-Bayes Generalization Bounds via Shifted Rademacher Processes

**Jun Yang**[*]
Department of Statistical Sciences
University of Toronto,
Vector Institute
jun@utstat.toronto.edu

**Shengyang Sun**[*]
Department of Computer Science
University of Toronto,
Vector Institute
ssy@cs.toronto.edu

**Daniel M. Roy**
Department of Statistical Sciences
University of Toronto,
Vector Institute
droy@utstat.toronto.edu

## Abstract

The developments of Rademacher complexity and PAC-Bayesian theory have been largely independent. One exception is the PAC-Bayes theorem of Kakade, Sridharan, and Tewari [21], which is established via Rademacher complexity theory by viewing Gibbs classifiers as linear operators. The goal of this paper is to extend this bridge between Rademacher complexity and state-of-the-art PAC-Bayesian theory. We first demonstrate that one can match the fast rate of Catoni's PAC-Bayes bounds [8] using shifted Rademacher processes [27, 43, 44]. We then derive a new fast-rate PAC-Bayes bound in terms of the "flatness" of the empirical risk surface on which the posterior concentrates. Our analysis establishes a new framework for deriving fast-rate PAC-Bayes bounds and yields new insights on PAC-Bayesian theory.

## 1 Introduction

PAC-Bayes theory [33, 38] was developed to provide *probably approximately correct* (PAC) guarantees for supervised learning algorithms whose outputs can be expressed as a weighted majority vote. Its uses have expanded considerably since [3, 6, 14, 17, 18, 28, 39, 40]. See [12, 25, 32] for gentle introductions. Indeed, there has been a surge of interest and work in PAC-Bayes theory and its application to large-scale neural networks, especially towards studying generalization in overparametrized neural networks trained by variants of gradient descent [9–11, 30, 36, 37].

PAC-Bayes bounds are one of several tools available for the study of the generalization and risk properties of learning algorithms. One advantage of the PAC-Bayes framework is its ease of use: one can obtain high-probability risk bounds for arbitrary ("posterior") Gibbs classifiers provided one can compute or bound relative entropies with respect to some fixed ("prior") Gibbs classifier. Another tool for studying generalization is Rademacher complexity, a distribution-dependent complexity measure for classes of real-valued functions [4, 5, 23, 29, 34, 44].

The literature on PAC-Bayes bounds and bounds based on Rademacher complexity are essentially disjoint. One point of contact is the work of Kakade, Sridharan, and Tewari [21], which builds the first bridge between PAC-Bayes theory and Rademacher complexity. By viewing Gibbs classifiers as

---

[*]These authors contributed equally.

linear operators and relative entropy as a strictly convex regularizer, they were able to use their general Rademacher complexity bounds on strictly convex linear classes to develop a slightly sharper version of McAllester's PAC-Bayes bound [33]. This result offers new insight on PAC-Bayes theory, including potential roles for data-dependent complexity estimates and stability. However, even within the PAC-Bayes community, this result is relatively unknown.

While the PAC-Bayes bound established by Kakade, Sridharan, and Tewari improves on McAllester's bound, it still converges at a slow $1/\sqrt{m}$ rate, where $m$ denotes the number of data used to form the empirical risk estimate. This observation raises the question of whether one can match state-of-the-art PAC-Bayes bounds via a Rademacher-process argument. In particular, can one match Catoni's bound [8, Thm. 1.2.6], which can obtain a fast $1/m$ rate of convergence?

There is an extensive literature on the problem of obtaining fast $1/m$ rates of convergence for the generalization error of (approximate) empirical risk minimization (ERM). Available approaches include the use of local Rademacher complexity [4, 22], shifted empirical processes [27], offset Rademacher complexities [29], and local empirical entropy [44]. See also [16, 19, 20, 26, 31, 35] and [13] for an extensive survey. To date, these techniques have not been connected to PAC-Bayesian theory, which presents the opportunity to obtain new PAC-Bayes theory for ERM.

## 1.1 Contributions

In this paper, we extend the bridge between Rademacher process theory and PAC-Bayes theory by constructing new bounds using Rademacher process techniques. Among our contributions:

i) We show how to recover Catoni's fast-rate PAC-Bayes bound [8], up to constants, using tail bounds on shifted Rademacher processes, which are special cases of shifted empirical processes [27, 43, 44]; See Section 3.

ii) We derive a new fast-rate PAC-Bayes bound, building on our shifted-Rademacher-process approach. This bound is determined by the "flatness" of the empirical risk surface on which the posterior Gibbs classifier concentrates. The notion of "flatness" is inspired by the proposal by Dziugaite and Roy [9] to formalize the empirical connection between "flat minima" and generalization using PAC-Bayes bounds; See Section 4.

iii) More generally, we introduce a new approach to derive fast-rate PAC-Bayes bounds and, in turn, offer new insight on PAC-Bayesian theory.

## 2 Background

Let $\mathscr{D}$ be an unknown distribution over a space $\mathscr{Z}$ of labeled examples, and let $\mathscr{H}$ be a hypothesis class. Relative to a binary loss function $\ell : \mathscr{H} \times \mathscr{Z} \to \{0,1\}$, we define the associated loss class $\mathscr{F} := \{\ell(h, \cdot) : h \in \mathscr{H}\}$ of functions from $\mathscr{Z} \to \{0,1\}$, each associated to one or more hypotheses. Let $\mathscr{L}_{\mathscr{D}}(f) := \mathbb{E}_{z \sim \mathscr{D}} f(z)$ denote the expected loss, i.e., risk, of every hypothesis associated to $f$. Let $S = (z_1, \cdots, z_m) \sim \mathscr{D}^m$ be a sequence of i.i.d. random variables. Let $\hat{\mathscr{L}}_S(f) = \frac{1}{m} \sum_{i=1}^m f(z_i)$ denote the empirical risk of every hypothesis associated to $f$.

We will be primarily interested in Gibbs classifiers, i.e., distributions $P$ on $\mathscr{F}$ which are interpreted as randomized classifiers that classify each new example according to a hypothesis drawn independently from $P$. (It is more common to work with distributions over $\mathscr{H}$, but these lead to looser results.) For a Gibbs classifier $P$ and labeled example $z \in \mathscr{Z}$, let $\mathbb{E}_P f(z) = \mathbb{E}_{f \sim P}[f(z)]$ be the expected loss $P$ suffers when labeling $z$. For Gibbs classifiers, the (expected) risk is defined to be $\mathscr{L}_{\mathscr{D}}(P) := \mathbb{E}_{f \sim P} \mathscr{L}_{\mathscr{D}}(f) = \mathbb{E}_{z \sim \mathscr{D}} \mathbb{E}_P f(z)$. The (expected) empirical risk is $\hat{\mathscr{L}}_S(P) := \mathbb{E}_{f \sim P} \hat{\mathscr{L}}_S(f) = \frac{1}{m} \sum_{i=1}^m \mathbb{E}_P f(z_i)$.

## 2.1 PAC-Bayes

The PAC-Bayes framework [33] provides data-dependent generalization guarantees for Gibbs classifiers. Each bound is specified in terms of a Gibbs classifier $P$ called the *prior*, as it must be independent of the training sample. The bound then holds for all *posterior* distributions, i.e., Gibbs classifiers that may be defined in terms of the training sample.

**Theorem 2.1** (PAC-Bayes [33])**.** *For any prior distribution $P$ over $\mathscr{F}$, for any $\delta \in (0,1)$, with probability at least $1-\delta$ over draws of training data $S \sim \mathscr{D}^m$, for all distributions $Q$ over $\mathscr{F}$,*

$$\mathscr{L}_{\mathscr{D}}(Q) \leq \hat{\mathscr{L}}_S(Q) + \sqrt{\frac{\mathrm{KL}\,(Q||P) + \log\frac{m}{\delta}}{2(m-1)}}. \tag{1}$$

Note in Theorem 2.1, the generalization bound scales as $\mathscr{O}(m^{-\frac{1}{2}})$. Catoni [8] presents a *fast rate* PAC-Bayesian bound, in which the generalization bound scales as $\mathscr{O}(m^{-1})$.

**Theorem 2.2** (Fast-Rate PAC-Bayes [8, Thm 1.2.6])**.** *For any prior distribution $P$ over $\mathscr{F}$, for any $\delta \in (0,1)$ and $C > 0$, with probability at least $1-\delta$ over draws of training data $S \sim \mathscr{D}^m$, for all distributions $Q$ over $\mathscr{F}$,*

$$\mathscr{L}_{\mathscr{D}}(Q) \leq \frac{1}{1 - e^{-C}} \left[ C\hat{\mathscr{L}}_S(Q) + \frac{\mathrm{KL}\,(Q||P) + \log\frac{1}{\delta}}{m} \right]. \tag{2}$$

Because the constant $C/(1 - e^{-C}) > 1$ holds for any $C > 0$, the generalization bound in Theorem 2.2 will always be bounded below by the empirical risk. Usually for a well-trained distribution $Q$ over training set, the empirical risk $\hat{\mathscr{L}}_S(Q)$ is small, therefore the generalization bound is dominated by the KL term. Compared to the standard PAC-Bayes bound in Theorem 2.1, where the KL term decreases at a rate $\mathscr{O}(m^{-\frac{1}{2}})$, the KL term of Catoni's bound decreases at a rate $\mathscr{O}(m^{-1})$. For this reason, we say that Catoni's bound achieves a *fast* rate of convergence. Note that fast-rate bounds can lead to much tighter bounds. Of course, $C/(1 - e^{-C}) \to 1$ as $C \to 0$, but, in that limit, the constants ignored in the asymptotic rate $\mathscr{O}(m^{-1})$ degrade. (See [28] for more discussion.)

## 2.2 Rademacher Viewpoint

Fix a prior Gibbs classifier $P$ on $\mathscr{F}$. Then, for measurable functions $g, h$, consider the inner product $\langle g, h \rangle = \int g(f)h(f)P(\mathrm{d}f)$. The key observation of Kakade, Sridharan, and Tewari is that one can view $\mathscr{L}_{\mathscr{D}}(Q)$ (resp., $\hat{\mathscr{L}}_S(Q)$) as the inner product $\langle \mathrm{d}Q/\mathrm{d}P, \mathscr{L}_{\mathscr{D}}(\cdot) \rangle$ (resp., $\langle \mathrm{d}Q/\mathrm{d}P, \hat{\mathscr{L}}_S(\cdot) \rangle$) between the posterior $Q$, represented by its Radon–Nikodym derivative with $P$, and the risk (resp., empirical risk), viewed as measurable function on $\mathscr{F}$. Thus, Gibbs classifiers can be viewed as linear predictors. Using their distribution-independent bounds on the Rademacher complexity of certain classes of linear predictors, Kakade, Sridharan, and Tewari [21] derive a PAC-Bayes bound similar to Theorem 2.1. We refer to this as the "Rademacher viewpoint" on PAC-Bayes.

We now summarize their argument in more detail. Let $\mathscr{Q}(\kappa) := \{Q : \mathrm{KL}\,(Q||P) \leq \kappa\}$. One can follow the classical steps for controlling the generalization error uniformly over $\mathscr{Q}(\kappa)$ using Rademacher complexity. Their first step is to connect $\sup_{Q \in \mathscr{Q}(\kappa)} \left[ \mathscr{L}_{\mathscr{D}}(Q) - \hat{\mathscr{L}}_S(Q) \right]$ to $\mathbb{E}_S \sup_{Q \in \mathscr{Q}(\kappa)} \left[ \mathscr{L}_{\mathscr{D}}(Q) - \hat{\mathscr{L}}_S(Q) \right]$ by the bounded difference inequality (McDiarmid's inequality). In particular, with probability at least $1 - \delta$,

$$\sup_{Q \in \mathscr{Q}(\kappa)} \left[ \mathscr{L}_{\mathscr{D}}(Q) - \hat{\mathscr{L}}_S(Q) \right] \leq \mathbb{E}_S \sup_{Q \in \mathscr{Q}(\kappa)} \left[ \mathscr{L}_{\mathscr{D}}(Q) - \hat{\mathscr{L}}_S(Q) \right] + \sqrt{\frac{\log(1/\delta)}{m}}. \tag{3}$$

Then they apply a symmetrization argument to obtain an upper bound in terms of Rademacher complexity [5]. In particular, recalling that $S = (z_1, \cdots, z_m)$ is our training data,

$$\mathbb{E}_S \sup_{Q \in \mathscr{Q}(\kappa)} \left[ \mathscr{L}_{\mathscr{D}}(Q) - \hat{\mathscr{L}}_S(Q) \right] \leq 2\mathbb{E}_S \mathbb{E}_{\epsilon} \sup_{Q \in \mathscr{Q}(\kappa)} \left[ \frac{1}{m} \sum_{i=1}^m \varepsilon_i \mathbb{E}_Q f(z_i) \right], \tag{4}$$

where $\{\varepsilon_i\}$ are i.i.d. Rademacher random variables, i.e., $\mathbb{P}(\varepsilon_i = +1) = \mathbb{P}(\varepsilon_i = -1) = 1/2$. Their last step is to bound the Rademacher complexity $\mathbb{E}_S \mathbb{E}_{\epsilon} \sup_{Q \in \mathscr{Q}(\kappa)} \left[ \frac{1}{m} \sum_{i=1}^m \varepsilon_i \mathbb{E}_Q f(z_i) \right]$, which can be seen as the Rademacher complexity of a linear class with a (strongly) convex constraint [21]. According to [21], the Rademacher complexity in Eq. (4) is of order $\sqrt{\kappa/m}$, which eventually leads to a term of order $\sqrt{\mathrm{KL}\,(Q||P)/m}$ after applying a union bound argument on $\kappa$.

In the end, using the above arguments and their sharp bounds on the Rademacher and Gaussian complexities of (constrained) linear classes [21, Thm. 1], Kakade, Sridharan, and Tewari obtain the

following PAC-Bayes bound [21, Cor. 8]: for every prior $P$ over $\mathscr{F}$, with probability at least $1 - \delta$ over draws of training data $S \sim \mathscr{D}^m$, for all distribution $Q$ over $\mathscr{F}$,

$$\mathscr{L}_{\mathscr{D}}(Q) \leq \hat{\mathscr{L}}_S(Q) + 4.5\sqrt{\frac{\max\{\mathrm{KL}(Q||P), 2\}}{m}} + \sqrt{\frac{\log(1/\delta)}{m}}. \tag{5}$$

Note that this PAC-Bayes bound has a slow rate of $\sqrt{1/m}$, but it slightly improves the rate in the term $\sqrt{\log(m/\delta)/m}$ of McAllester's bound [33] to $\sqrt{\log(1/\delta)/m}$.

Since McAllester's bound is far from the state-of-art in PAC-Bayesian theory, this raises the question whether one can extend the "Rademacher viewpoint" of PAC-Bayes to derive more advanced bounds, such as one matching the fast rate of Catoni's bound.

## 3 Extending the Rademacher Viewpoint

There are at least two difficulties in the "Rademacher viewpoint" that prevent fast rates. First, if we connect the generalization error to Rademacher complexity using the bounded difference inequality, a slow rate term $\sqrt{\log(1/\delta)/m}$ will appear. Second, as is shown by Kakade, Sridharan, and Tewari [21], the standard Rademacher complexity of (constraint) linear classes leads to an upper bound with a slow rate of order $\mathscr{O}(\sqrt{\mathrm{KL}(Q||P)/m})$. Therefore, in order to derive fast rate PAC-Bayes bounds, we need to extend the "Rademacher viewpoint".

In order to obtain fast rates, we work with so-called shifted Rademacher processes, i.e., processes of the form $\{\frac{1}{m}\sum_{i=1}^m \varepsilon_i' f(z_i)\}_{f \in \mathscr{F}}$ where the variables $\{\varepsilon_i'\}$ are independent from $S$, i.i.d., and take two values with equal probability. (These shifted Rademacher variables, $\{\varepsilon_i'\}$, are not necessarily zero mean. When they take values in $\{\pm 1\}$, we obtain a standard Rademacher process.) Shifted Rademacher processes are examples of shifted empirical processes [27, 43, 44].

Recall that Rademacher complexity is the *expected value* of the supremum of Rademacher processes over a class [5]. In order to get a fast rate, we connect the *tail probabilities* of the supremum of the generalization error to the *tail probabilities* of shifted Rademacher processes via a symmetrization-in-deviation argument instead of the symmetrization-in-expectation argument. The key is that we can avoid using the bounded difference inequality by bounding the deviation. This removes the slow rate term of $\sqrt{\log(1/\delta)/m}$. It remains to bound the deviation of shifted Rademacher processes to get a fast rate bound of order $\mathscr{O}(\mathrm{KL}(Q||P)/m)$.

In the following, we demonstrate how the extended "Rademacher viewpoint" via shifted Rademacher processes can be applied to derive a fast rate PAC-Bayes bound that matches the fast rate of Catoni's bound. Note that, since $C/(1 - e^{-C}) > 1$ for fixed $C > 0$ in Catoni's bound in Eq. (2), we can write $C/(1 - e^{-C}) = 1 + c$ for some constant $c > 0$. Furthermore, note that our goal in this section is not to derive new PAC-Bayes bounds. Therefore, we do not make attempts to optimize the constants.

**Proposition 3.1** (Matching Catoni's Fast Rate via Shifted Rademacher Processes). *For any given $c > 0$ and prior $P$ over $\mathscr{F}$, there exists constants $C_1$, $C_2$, and $C_3$ such that, with probability at least $1 - \delta$, for all distributions $Q$ over $\mathscr{F}$*

$$\mathscr{L}_{\mathscr{D}}(Q) \leq (1 + c)\hat{\mathscr{L}}_S(Q) + C_1 \frac{\mathrm{KL}(Q||P)}{m} + C_2 \frac{\log \frac{1}{\delta}}{m} + C_3 \frac{1}{m}. \tag{6}$$

*Outline of the proof.* We wish to emphasize two key differences from traditional machinery for deriving Rademacher-complexity-based generalization bounds. The complete proof is given in Appendix A.1.

Fix $P$ and let $\mathscr{Q}(\kappa) := \{Q : \mathrm{KL}(Q||P) \leq \kappa\}$ be defined as in Section 2.2. Rather than control $\sup_{Q \in \mathscr{Q}(\kappa)} \left[\mathscr{L}_{\mathscr{D}}(Q) - \hat{\mathscr{L}}_S(Q)\right]$ in terms of its *expectation* via the bounded difference inequality and Rademacher complexity, we bound the *tail/deviation* of $\sup_{Q \in \mathscr{Q}(\kappa)} \left[\mathscr{L}_{\mathscr{D}}(Q) - (1 + c)\hat{\mathscr{L}}_S(Q)\right]$, thus avoiding the use of the bounded differences inequality altogether. In particular, we can obtain fast rates by bounding the tail in terms of tail of supremum of shifted Rademacher processes [27, 43, 44].

Define $\mathscr{G}_\kappa := \{\mathbb{E}_Q f(\cdot) : Q \in \mathscr{Q}(\kappa)\}$ and, by an abuse of notation, let $\mathscr{L}_\mathscr{D}(g)$ denote $\mathbb{E}_{z\sim\mathscr{D}}[g(z)]$. Then we can write $\sup_{Q\in\mathscr{Q}(\kappa)}\left[\mathscr{L}_\mathscr{D}(Q) - (1+c)\hat{\mathscr{L}}_S(Q)\right]$ as $\sup_{g\in\mathscr{G}_\kappa}\left[\mathscr{L}_\mathscr{D}(g) - (1+c)\hat{\mathscr{L}}_S(g)\right]$. We start from bounding the tail probability $\mathbb{P}_S\left(\sup_{g\in\mathscr{G}_\kappa}\mathscr{L}_\mathscr{D}(g) - (1+c)\hat{\mathscr{L}}_S(g) \geq t\right)$. For fixed constants $c > c_2 > 0$, let $c' = \frac{c-c_2}{1+c_2}$ and $t' = \frac{t}{2(1+c_2)}$. Then, by [44, Cor. 1], we have

$$\mathbb{P}_S\left(\sup_{g\in\mathscr{G}_\kappa}\mathscr{L}_\mathscr{D}(g) - (1+c)\hat{\mathscr{L}}_S(g) \geq t\right) \leq 4\mathbb{P}_{S,\varepsilon}\left(\sup_{g\in\mathscr{G}_\kappa}\left[\frac{1+\frac{c'}{2}}{m}\sum_{i=1}^m\left(\varepsilon_i - \frac{c'}{2+c'}\right)g(z_i)\right] \geq \frac{t'}{2}\right). \quad (7)$$

Letting $\varepsilon_i' := \varepsilon_i - \frac{c'}{2+c'}$, one can see that $\{\varepsilon_i'\}$ are i.i.d. "shifted" Rademacher random variables with mean $-\frac{c'}{2+c'}$. For any $g \in \mathscr{G}_\kappa$, there exists $Q \in \mathscr{Q}(\kappa)$ such that

$$\frac{1}{m}\sum_{i=1}^m\varepsilon_i'g(z_i) = \frac{1}{m}\sum_{i=1}^m\varepsilon_i'\mathbb{E}_Q f(z_i) = \mathbb{E}_Q\left[\frac{1}{m}\sum_{i=1}^m\varepsilon_i'f(z_i)\right], \quad (8)$$

which can be viewed as a linear function of $Q$. Further, it can be verified that the set $\mathscr{Q}(\kappa)$ is (strongly) convex. Therefore, $\sup_{Q\in\mathscr{Q}(\kappa)}\frac{1}{m}\sum_{i=1}^m\varepsilon_i'\mathbb{E}_Q f(z_i)$ is a convex optimization problem. By duality [7, Chp. 5], and, in this particular case, the Legendre transform of Kullback–Leibler divergence (see, e.g., [18]), we have

$$\sup_{g\in\mathscr{G}_\kappa}\frac{1}{m}\sum_{i=1}^m\varepsilon_i'g(z_i) = \sup_{Q\in\mathscr{Q}(\kappa)}\frac{1}{m}\sum_{i=1}^m\varepsilon_i'\mathbb{E}_Q f(z_i) = \inf_{\lambda>0}\left\{\frac{\kappa}{\lambda} + \frac{1}{\lambda}\log\mathbb{E}_P\left[\exp\left(\frac{\lambda}{m}\sum_{i=1}^m\varepsilon_i'f(z_i)\right)\right]\right\}. \quad (9)$$

Combining the shifted symmetrization in deviation in Eq. (7) and the dual problem in Eq. (9), Markov's inequality yields, for every $\lambda > 0$,

$$\mathbb{P}_S\left(\sup_{Q\in\mathscr{Q}(\kappa)}\mathbb{E}_Q[\mathscr{L}_\mathscr{D}(f) - (1+c)\hat{\mathscr{L}}_S(f)] \geq t\right) \leq 4e^{\kappa - \frac{\lambda t'}{2+c'}}\mathbb{E}_S\mathbb{E}_\varepsilon\mathbb{E}_P\left[\exp\left(\frac{\lambda}{m}\sum_{i=1}^m\varepsilon_i'f(z_i)\right)\right]. \quad (10)$$

We then exploit the shifted property of $\varepsilon_i'$ to bound the expectation term on the right-hand side and obtain fast rates. In particular, we show that, so long as $k \geq \frac{\log\cosh(\lambda/m)}{\lambda/m}$,

$$\mathbb{E}_P\mathbb{E}_S\mathbb{E}_\epsilon\left[\exp\left(\frac{\lambda}{m}\sum_{i=1}^m(\varepsilon_i - k)f(z_i)\right)\right] \leq 1. \quad (11)$$

In our case, $k = \frac{c'}{2+c'}$, which leads to constraints relating $\lambda$, $c$, and $c_2$. In particular, when $c = 0$, the required condition for the above result, $k \geq \frac{\log\cosh(\lambda/m)}{\lambda/m}$, does not hold. Therefore, this approach obtains fast rates only if $c > 0$, i.e., if we shift. Combing Eqs. (10) and (11), there exists a constant $C'$, depending only on $c$, $c_2$ and $\delta$, such that, with probability at least $1 - \delta$,

$$\sup_{Q\in\mathscr{Q}(\kappa)}\mathbb{E}_Q[\mathscr{L}_\mathscr{D}(f) - (1+c)\hat{\mathscr{L}}_S(f)] \leq \frac{C'}{m}(\kappa + \log(4/\delta)). \quad (12)$$

Finally, we may apply the same union-bound argument as in the proof of [21, Cor. 7] in order to cover all possible values of $\kappa$. This completes the proof. □

## 4 New Fast Rate PAC-Bayes Bound based on "Flatness"

The extended " Rademacher viewpoint" of PAC-Bayes provides a new approach for deriving fast-rate PAC-Bayes bounds. In this section, we demonstrate the use of shifted Rademacher processes to derive a new fast-rate PAC-Bayes bound using a notion of "flatness". This notion is inspired by the proposal by Dziugaite and Roy [9] to formalize the empirical connection between "flat minima" and generalization using PAC-Bayes bounds, and, in particular, posterior distributions which concentrate in these "flat minima".

**Definition 4.1** (Notion of "Flatness"). For given $h \in [0,1]$, the "$h$-flatness" of $Q$ (w.r.t. $S$) is

$$\frac{1}{m}\sum_{i=1}^m\mathbb{E}_Q[f(z_i) - (1+h)\mathbb{E}_Q f(z_i)]^2. \quad (13)$$

One way to understand this new notion is to observe that, under zero–one loss, $h$-flatness can be written as the difference between the empirical risk and the quadratic empirical risk:

$$\frac{1}{m}\sum_{i=1}^{m}\mathbb{E}_Q[f(z_i)-(1+h)\mathbb{E}_Qf(z_i)]^2 = \hat{\mathscr{L}}_S(Q) - \frac{1-h^2}{m}\sum_{i=1}^{m}(\mathbb{E}_Qf(z_i))^2. \qquad (14)$$

Note that, for $[0,1]$-valued (bounded) loss, equality is replaced by an inequality: the r.h.s. is an upper bound of the l.h.s.

*Remark* 4.2. To see that optimizing $h$-flatness prefers "flat minima", consider the following simplified case: Call a posterior $Q$ "completely flat" if $f=g$ on $S$ a.s., when $f,g \sim Q$. It can be verified that, if the posterior is "completely flat", then under the zero–one loss, the "$h$-flatness" is $h^2\hat{\mathscr{L}}_S(Q)$. That is, given a "completely flat" posterior, the "$h$-flatness" goes to zero as $h \to 0$. For $h > 0$, the "$h$-flatness" is zero when $Q$ is "completely flat" and $\hat{\mathscr{L}}_S(Q)=0$. ◁

The following PAC-Bayes theorem establishes favorable bounds for $h$-flat posteriors:

**Theorem 4.3** (Fast Rate PAC-Bayes using "Flatness"). *For any given $c > 0$ and $h \in (0,1)$, with probability at least $1-\delta$ over random draws of training set $S \sim \mathscr{D}^m$, for all distributions $Q$ over $\mathscr{F}$,*

$$\mathscr{L}_{\mathscr{D}}(Q) \leq \hat{\mathscr{L}}_S(Q) + \frac{c}{m}\sum_{i=1}^{m}\mathbb{E}_Q[f(z_i)-(1+h)\mathbb{E}_Qf(z_i)]^2 + \frac{4}{Cm}\left[3\mathrm{KL}(Q||P)+\log\frac{1}{\delta}+5\right], \qquad (15)$$

*where $C = \frac{2h^4c}{1+16h^2c}$.*

This bound can be tighter than Catoni's bound under certain conditions. We delay the comparison with Catoni's bound to Section 4.1. We now give an outline of the proof of Theorem 4.3, highlighting the technical differences from the proof of Proposition 3.1. The complete proof is given in Appendix A.2.

*Outline of the proof of Theorem 4.3.* By Eq. (14), we can write

$$\mathbb{E}_Q\mathscr{L}_{\mathscr{D}}(f) - \hat{\mathscr{L}}_S(Q) - \frac{c}{m}\sum_{i=1}^{m}\mathbb{E}_Q[f(z_i)-(1+h)\mathbb{E}_Qf(z_i)]^2$$

$$= \mathscr{L}_{\mathscr{D}}(Q) - (1+c)\hat{\mathscr{L}}_S(Q) + \frac{c(1-h^2)}{m}\sum_{i=1}^{m}(\mathbb{E}_Qf(z_i))^2. \qquad (16)$$

There are at least two new challenges compared with the proof of Proposition 3.1. First, the shifted symmetrization in Eq. (7) cannot be applied because of the existence of the quadratic term $\frac{c(1-h^2)}{m}\sum_{i=1}^{m}(\mathbb{E}_Qf(z_i))^2$. This means we need to derive a new shifted symmetrization involving the quadratic term. Second, the quadratic term $\frac{c(1-h^2)}{m}\sum_{i=1}^{m}(\mathbb{E}_Qf(z_i))^2$ cannot be seen as a linear function of $Q$. Therefore, some technical arguments are required in order to apply the Legendre transform of Kullback–Leibler divergence.

First, we derive a new shifted symmetrization which involves quadratic terms. The proof is inspired by an argument due to Zhivotovskiy and Hanneke [44]. The result extends [44, Cor. 1], which is recovered as a special case when $h=1$. For $\kappa > 0$, recall that we have defined $\mathscr{Q}(\kappa) = \{Q : \mathrm{KL}(Q||P) \leq \kappa\}$ and $\mathscr{G}_\kappa = \{\mathbb{E}_Qf(\cdot) : Q \in \mathscr{Q}(\kappa)\}$. Then for any $g \in \mathscr{G}_\kappa$, there exists a $Q \in \mathscr{Q}(\kappa)$ such that $g = \mathbb{E}_Qf(\cdot)$. We can first show a tail bound that for any given $c_2 > 0$ and $g \in \mathscr{G}_\kappa$, if $t \geq \frac{(1+c_2)(1+c_2h^2)}{mc_2h^2}$, then

$$\mathbb{P}_S\left(\mathscr{L}_{\mathscr{D}}(g) - (1+c_2)\hat{\mathscr{L}}_S(g) + c_2(1-h^2)\hat{\mathscr{L}}_S(g^2) \geq \frac{t}{2}\right) \leq \frac{1}{2}. \qquad (17)$$

Then, consider another independent random data set $S' = \{z'_1, \ldots, z'_m\} \in \mathscr{D}^m$. For $c > c_2$, by taking the difference of $\mathscr{L}_{\mathscr{D}}(g) - (1+c)\hat{\mathscr{L}}_S(g) + c(1-h^2)\hat{\mathscr{L}}_S(g^2)$ and $\mathscr{L}_{\mathscr{D}}(g) - (1+c_2)\hat{\mathscr{L}}_{S'}(g) + c_2(1-$

$h^2)\hat{\mathscr{L}}_{S'}(g^2)$ and using Eq. (17), we obtain

$$\frac{1}{4}\mathbb{P}_S\left(\sup_{g\in\mathscr{G}_\kappa}\mathscr{L}_{\mathscr{D}}(g)-(1+c)\hat{\mathscr{L}}_S(g)+c(1-h^2)\hat{\mathscr{L}}_S(g^2)\geq t\right) \tag{18}$$

$$\leq\frac{1}{2}\mathbb{P}_{S,S'}\left(\sup_{g\in\mathscr{G}_\kappa}(1+c_2)\hat{\mathscr{L}}_{S'}(g)-c_2(1-h^2)\hat{\mathscr{L}}_{S'}(g^2)-(1+c)\hat{\mathscr{L}}_S(g)+c(1-h^2)\hat{\mathscr{L}}_S(g^2)\geq\frac{t}{2}\right). \tag{19}$$

Now by writing $(1+c_2)\hat{\mathscr{L}}_{S'}(g)-c_2(1-h^2)\hat{\mathscr{L}}_{S'}(g^2)-(1+c)\hat{\mathscr{L}}_S(g)+c(1-h^2)\hat{\mathscr{L}}_S(g^2)$ as

$$(1+\frac{c+c_2}{2})\left(\hat{\mathscr{L}}_{S'}(g)-\hat{\mathscr{L}}_S(g)\right)-\frac{c+c_2}{2}(1-h^2)\left(\hat{\mathscr{L}}_{S'}(g^2)-\hat{\mathscr{L}}_S(g^2)\right)$$

$$-\frac{c-c_2}{2}\hat{\mathscr{L}}_S\left(g-(1-h^2)g^2\right)-\frac{c-c_2}{2}\hat{\mathscr{L}}_{S'}\left(g-(1-h^2)g^2\right), \tag{20}$$

one can apply the symmetrization argument to get

$$\frac{1}{4}\mathbb{P}_S\left(\sup_{g\in\mathscr{G}_\kappa}\mathscr{L}_{\mathscr{D}}(g)-(1+c)\hat{\mathscr{L}}_S(g)+c(1-h^2)\hat{\mathscr{L}}_S(g^2)\geq t\right)$$

$$\leq\mathbb{P}_{S,\epsilon}\left(\sup_{g\in\mathscr{G}_\kappa}\left[\frac{1}{m}\sum_{i=1}^m\varepsilon_i\left((1+c')\,g(z_i)-c'(1-h^2)g^2(z_i)\right)-c''\hat{\mathscr{L}}_S\left(g-(1-h^2)g^2\right)\right]\geq\frac{t}{4}\right), \tag{21}$$

where $c'=\frac{c+c_2}{2},c''=\frac{c-c_2}{2}$. Therefore, we have derived the new shifted symmetrization in deviation involving a quadratic term.

Recalling the definition of $\mathscr{G}_\kappa$, we have

$$\sup_{g\in\mathscr{G}_\kappa}\frac{1}{m}\sum_{i=1}^m\varepsilon_i\left((1+c')\,g(z_i)-c'(1-h^2)g(z_i)^2\right)-c''\hat{\mathscr{L}}_S(g-(1-h^2)g^2)$$

$$=\sup_{Q\in\mathscr{Q}(\kappa)}\frac{1}{m}\sum_{i=1}^m[\varepsilon_i\left(1+c'\right)-c'']\mathbb{E}_Qf(z_i)-\left[\varepsilon_ic'-c''\right](1-h^2)[\mathbb{E}_Qf(z_i)]^2. \tag{22}$$

Note that there are two shifted Rademacher random variables $\varepsilon_i\left(1+c'\right)-c''$ and $\varepsilon_ic'-c''$, which not only involve a shift term $-c''$ but also scale terms $(1+c')$ and $c'$, respectively. Furthermore, the term $[\mathbb{E}_Qf(z_i)]^2$ cannot be seen as a linear function of $Q$. This prevents the use of the key argument in [21] to formulate an upper bound using Rademacher complexities of constrained linear classes by considering the generalization error as a linear function of $Q$.

In order to sidestep this obstruction, define $\epsilon:=\{\varepsilon_i\}_{i=1}^m,\mathbf{z}:=\{z_i\}_{i=1}^m$ and suppose $\hat{Q}(\epsilon,\mathbf{z})$ achieves the supremum above. (If the supremum cannot be achieved, one can use a carefully chosen sequence of $\{\hat{Q}_i(\epsilon,\mathbf{z})\}$ to prove the same statement as the supremum can be approximated arbitrarily closely.) The following inequality then holds:

$$\sup_{Q\in\mathscr{Q}(\kappa)}\frac{1}{m}\sum_{i=1}^m[\varepsilon_i\left(1+c'\right)-c'']\mathbb{E}_Qf(z_i)-\left[\varepsilon_ic'-c''\right](1-h^2)[\mathbb{E}_Qf(z_i)]^2$$

$$\leq\sup_{Q\in\mathscr{Q}(\kappa)}\frac{1}{m}\sum_{i=1}^m[\varepsilon_i\left(1+c'\right)-c'']\mathbb{E}_Qf(z_i)-\left[\varepsilon_ic'-c''\right](1-h^2)\mathbb{E}_Qf(z_i)\mathbb{E}_{\hat{Q}(\epsilon,\mathbf{z})}f(z_i). \tag{23}$$

To see this, note that, on the one hand, if we plug in $Q=\hat{Q}(\epsilon,\mathbf{z})$ the inequality is tight; on the other hand, by definition, $Q=\hat{Q}(\epsilon,\mathbf{z})$ already achieves the supremum of the l.h.s. Note that the r.h.s. can be seen as a linear function of $Q$, because $\hat{Q}(\epsilon,\mathbf{z})$ is a random variable which does not depend on $Q$.

Let $\varepsilon_i'':=\varepsilon_ic'-c''=\varepsilon_i\frac{c_1+c_2}{2}-\frac{c_1-c_2}{2}$. Then by keeping the term $\hat{Q}(\epsilon,\mathbf{z})$, one can apply the convex conjugate of relative entropy to get

$$\mathbb{P}\left[\sup_{Q\in\mathscr{Q}(\kappa)}\mathscr{L}_{\mathscr{D}}(Q)-(1+c)\hat{\mathscr{L}}_S(Q)+\frac{c(1-h^2)}{m}\sum_{i=1}^m(\mathbb{E}_Qf(z_i))^2\geq t\right]$$

$$\leq 4\exp\left(\kappa-\frac{\lambda t}{4}\right)\mathbb{E}_S\mathbb{E}_\epsilon\mathbb{E}_P\left[\exp\left(\frac{\lambda}{m}\sum_{i=1}^m f(z_i)\left[(\varepsilon_i+\varepsilon_i'')-\varepsilon_i''(1-h^2)\mathbb{E}_{\hat{Q}(\epsilon,\mathbf{z})}f(z_i)\right]\right)\right]. \tag{24}$$

Therefore, the problem turns to bounding the expectation of a function involving shifted Rademacher processes. Although the expectation looks quite complicated since it involves two scaled and shifted Rademacher variables as well as the unknown $\hat{Q}(\epsilon, \mathbf{z})$, fortunately, we are able to show that, for any random variables $Y_i \in [0, 1]$, we have

$$\mathbb{E}_S \mathbb{E}_\epsilon \mathbb{E}_P \left[ \exp \left( \frac{\lambda}{m} \sum_{i=1}^m f(z_i) \left[ (\varepsilon_i + \varepsilon_i'') - \varepsilon_i''(1 - h^2) Y_i \right] \right) \right] \leq 1, \tag{25}$$

if $h \in (0, 1], 1 > h^2 c > c_2 > 0$ and $0 < \frac{\lambda}{m} < C = \frac{h^2 c - c_2}{2(1 + h^2 c)(1 + c_2)}$. This result removes the term $\hat{Q}(\epsilon, \mathbf{z})$ by letting $Y_i = \mathbb{E}_{\hat{Q}(\epsilon, \mathbf{z})} f(z_i)$. Finally, we combine different values of $\kappa$ by a union bound argument similar to the proof of Proposition 3.1 to complete the proof. □

## 4.1 Comparison with Catoni's Bound

As we have shown in Proposition 3.1, using shifted Rademacher processes, we can match Catoni's fast-rate PAC-Bayesian bound (Theorem 2.2) up to constants. We have also presented a new fast-rate PAC-Bayes bound based on "flatness". Although both our bound and Catoni's bound show fast $\mathcal{O}(m^{-1})$ rates of convergence, our bound can exploit flatness in the posterior distribution.

In particular, our PAC-Bayes bound based on flatness (Eq. (15)) can be much tighter than Catoni's bound (Eq. (6)) when the posterior is chosen to concentrate on a "flat minimum" where $\frac{c}{m} \sum_{i=1}^m \mathbb{E}_Q[f(z_i) - (1 + h)\mathbb{E}_Q f(z_i)]^2$ is very small yet $\hat{\mathcal{L}}_S(Q)$ is nonzero. It can be verified that the "flatness" term $\frac{c}{m} \sum_{i=1}^m \mathbb{E}_Q[f(z_i) - (1 + h)\mathbb{E}_Q f(z_i)]^2$ in Eq. (15) is smaller than the excess empirical risk term $c\hat{\mathcal{L}}_S(Q)$ when $\frac{1-h^2}{m} \sum_{i=1}^m (\mathbb{E}_Q f(z_i))^2$ is greater than 0, which is precisely when the empirical risk is greater than zero. (See Eq. (14).)

Based on this observation, we expect our bound to be tighter for sufficient flat posteriors, nonzero empirical risk, and sufficient training data. In order to see this, note that Catoni's bound has the form $(1 + c_c)\hat{\mathcal{L}}_S Q + \frac{\mathscr{C}_c}{m}(\text{KL}(Q\|P) + \log \frac{1}{\delta})$, while our bound based on Eq. (14) can be written $(1 + c_r)\hat{\mathcal{L}}_S Q - \frac{c_r(1-h^2)}{m} \sum_{i=1}^m (\mathbb{E}_Q f(z_i))^2 + \frac{\mathscr{C}_r}{m}(\text{KL}(Q\|P) + \log \frac{1}{\delta} + 1)$. Here $c_c, c_r$ inflate the empirical risk and $\mathscr{C}_c, \mathscr{C}_r$ are constants. Let $T_m$ be $\frac{c_r(1-h^2)}{m} \sum_{i=1}^m (\mathbb{E}_Q f(z_i))^2$. Note that $c_c$ and $c_r$ must be fixed before seeing the data. Assuming we equate the inflation of the empirical risk, i.e., $c_c = c_r$, the proposed bound is tighter than Catoni's bound provided $m > \frac{1}{T_m} \left( (\mathscr{C}_r - \mathscr{C}_c) \left( \text{KL}(Q\|P) + \log \frac{1}{\delta} \right) + \mathscr{C}_r \right)$. If $T_m$ converges to a positive number (a reasonable assumption), then our proposed bound will be tighter for sufficiently many samples. If we assume $c_c \neq c_r$, our bound can still be tighter than Catoni's bound under more involved conditions.

# 5 Related Work

There is a large literature on obtaining fast $1/m$ convergence rates for generalization error and excess risk using Rademacher processes and their generalizations [4, 22, 27, 29, 44]. As far as we know, this literature does not connect with the PAC-Bayesian literature. There do exist, however, PAC-Bayesian analyses for specific learning algorithms that achieve fast rates [2, 15, 24]. These specific analyses do not lead to general PAC-Bayes bounds, like those produced by Catoni [8].

Our new PAC-Bayes bound based on flatness bears a superficial resemblance to a number of bounds in the literature. However, our notion of flatness is *not* related to the variance of the randomized classifier caused by the randomness of the observed data. Therefore, our new bound is fundamentally different from existing PAC-Bayes bounds based on this type of variance [15, 24, 41].

For example, Tolstikhin and Seldin [41, Thm. 4] presents a generalization bound based on the "empirical variance", which is distinct from our "flatness". The "empirical variance" is $\mathbb{E}_Q \frac{1}{m} \sum_{i=1}^m [f(z_i) - \frac{1}{m} \sum_{i=1}^m f(z_i)]^2$, while our "flatness" is $\frac{1}{m} \sum_{i=1}^m \mathbb{E}_Q[f(z_i) - \mathbb{E}_Q f(z_i)]^2$. Note that it is possible for flatness to be zero, even when empirical variance is large.

To the best of our knowledge, the closest work to ours in the literature is that by Audibert [2]. The bound given in [2, Thm. 6.1] uses a notion similar to our "flatness". The bound is, however, not comparable with ours for several reasons: First, [2, Theorem 6.1] holds only for the particular

algorithm proposed by Audibert, and so it is not a general PAC-Bayes bound like ours. Second, our notion of "flatness" is empirical, while the "flatness" term in [2, Theorem 6.1] is defined by an expectation over the data distribution, which is often presumed unknown. Finally, the proof techniques used to establish [2, Theorem 6.1] are specialized to the proposed algorithm and not based on the use of Rademacher processes. Our proof techniques via shifted Rademacher processes provides a blueprint for other approaches to deriving fast-rate PAC-Bayes bounds.

Grünwald and Mehta [17] establish new excess risk bounds in terms of a novel complexity measure based on "luckiness" functions. In the setting of randomized classifiers, particular choices of luckiness functions can be related to PAC-Bayesian notions of complexity based on "priors". Indeed, in this setting, their complexity measure can be bounded in terms of a KL divergence, as in PAC-Bayesian bounds. In a setting with deterministic classifiers, the authors show that their complexity measure can be bounded in terms of Rademacher complexity. Thus, while their framework connects with both PAC-Bayesian and Rademacher-complexity bounds, it is not immediately clear whether it produces direct connections, as we have accomplished here. It is certainly interesting to consider whether our bounds can be achieved (or surpassed) by an appropriate use of their framework.

## 6 Conclusion

In this paper we exploit the connections between modern PAC-Bayesian theory and Rademacher complexities. Using shifted Rademacher processes [27, 43, 44], we derive a novel fast-rate PAC-Bayes bound that depends on the empirical "flatness" of the posterior. Our work provides new insights on PAC-Bayesian theory and opens up new avenues for developing stronger bounds.

It is worth highlighting some potentially interesting directions that may be worth further investigation:

We have "rederived" Catoni's bound via shifted Rademacher processes, up to constants. It is interesting to ask whether the Rademacher approach can dominate the direct PAC-Bayes bound. In the other direction, we have not derived our flatness bound via a direct PAC-Bayes approach. Whether this is possible and what it achieves might shed light on the relative strengths of these two distinct approaches to PAC-Bayes bounds. It may also be interesting to pursue PAC-Bayes bounds via some adaptation of Talagrand's concentration inequalities [42, Ch.3].

We have derived PAC-Bayes bounds for zero–one loss. While the extension to bounded loss is straightforward, the problem of extending our approach to unbounded loss relates to a growing body of work on this problem within the PAC-Bayesian framework. (See, for example, [1] and the references therein). Whether the Rademacher perspective is helpful or not in this regard is not clear at this point.

There has been a surge of interest in PAC-Bayes bounds and their application to the study of generalization in large-scale neural networks. One promising direction is to consider Rademacher-process techniques may aid in the development of PAC-Bayesian analyses of specific algorithms [2, 15, 24], especially in the case when the algorithms are related to large-scale neural networks trained by stochastic gradient descent [30, 36, 37].

It would be interesting to perform a careful empirical study of our flatness bound in the context of large-scale neural networks, in the vein of the work of Dziugaite and Roy [9]. Preliminary work suggests that the posteriors found by PAC-Bayes bound optimization are not flat in our sense. After some investigation, we believe the reason is that optimizing the PAC-Bayes bound results in underfitting, due in part to the distribution-independent prior. It would be interesting to compare various PAC-Bayes bounds under strict constraints on the empirical risk.

### Acknowledgments

We would like to also thank Peter Bartlett, Gintare Karolina Dziugaite, Roger Grosse, Yasaman Mahdaviyeh, Zacharie Naulet, and Sasha Rakhlin for helpful discussions. In particular, the authors would like to thank Sasha Rakhlin for introducing us to the work of Kakade, Sridharan, and Tewari [21]. The work benefitted also from constructive feedback from anonymous referees. JY was supported by an Alexander Graham Bell Canada Graduate Scholarship (NSERC CGS D), Ontario Graduate Scholarship (OGS), and Queen Elizabeth II Graduate Scholarship in Science and Technology (QEII-GSST). SS was supported by a Borealis AI Global Fellowship Award, Connaught New

Researcher Award, and Connaught Fellowship. DMR was supported by an NSERC Discovery Grant and Ontario Early Researcher Award.

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
