[Supplementary Material · fastpacbayes-full.pdf]

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

(1+c') - c''] \mathbb{E}_Q f(z_i) - [\varepsilon_i c' - c''] (1-h^2)[\mathbb{E}_Q f(z_i)]^2. \tag{22}$$

Note that there are two shifted Rademacher random variables $\varepsilon_i(1+c') - c''$ and $\varepsilon_i c' - c''$, which not only involve a shift term $-c''$ but also scale terms $(1+c')$ and $c'$, respectively. Furthermore, the term $[\mathbb{E}_Q f(z_i)]^2$ cannot be seen as a linear function of $Q$. This prevents the use of the key argument in [21] to formulate an upper bound using Rademacher complexities of constrained linear classes by considering the generalization error as a linear function of $Q$.

In order to sidestep this obstruction, define $\epsilon := \{\varepsilon_i\}_{i=1}^m, \mathbf{z} := \{z_i\}_{i=1}^m$ and suppose $\hat{Q}(\epsilon, \mathbf{z})$ achieves the supremum above. (If the supremum cannot be achieved, one can use a carefully chosen sequence of $\{\hat{Q}_i(\epsilon, \mathbf{z})\}$ to prove the same statement as the supremum can be approximated arbitrarily closely.) The following inequality then holds:

$$\sup_{Q \in \mathscr{Q}(\kappa)} \frac{1}{m} \sum_{i=1}^m [\varepsilon_i(1+c') - c''] \mathbb{E}_Q f(z_i) - [\varepsilon_i c' - c''] (1-h^2)[\mathbb{E}_Q f(

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

# A Proofs

## A.1 Proof of Proposition 3.1

To match Catoni's bound, we need to control $\sup_{f\in\mathscr{F}}\mathscr{L}_{\mathscr{D}}(f)-(1+c)\hat{\mathscr{L}}_S(f)$, given $c>0$. We apply an existing result due to Zhivotovskiy and Hanneke [44], which we quote here:

**Lemma A.1** (Shifted Symmetrization in deviation [44, Cor. 7]). *Fix constants $c>c_2>0$,*

$$\frac{1}{4}\mathbb{P}_S\left(\sup_{f\in\mathscr{F}}\mathscr{L}_{\mathscr{D}}(f)-(1+c)\hat{\mathscr{L}}_S(f)\geq t\right)$$

$$\leq \mathbb{P}_{S,\varepsilon}\left(\sup_{f\in\mathscr{F}}\left[\frac{1+c'/2}{m}\sum_{i=1}^{m}\left(\varepsilon_i-\frac{c'/2}{1+c'/2}\right)f(z_i)\right]\geq\frac{t'}{2}\right). \tag{26}$$

*where $c'=\frac{c-c_2}{1+c_2}, t'=\frac{t}{2(1+c_2)}$, and $\{\varepsilon_i\}$ are Rademacher random variables.*

Let $\varepsilon_i':=\varepsilon_i-\frac{c'}{2+c'}$. For $\kappa>0$, define $\mathscr{Q}(\kappa):=\{Q:\mathrm{KL}\,(Q||P)\leq\kappa\}$. Next, using convex conjugate of the Kullback–Leibler divergence (the change-measure inequality), for $\lambda>0$,

$$\sup_{Q\in\mathscr{Q}(\kappa)}\left[\frac{1}{m}\sum_{i=1}^{m}\varepsilon_i'\mathbb{E}_Q f(z_i)-\frac{1}{\lambda}\mathrm{KL}\,(Q||P)\right]\leq\frac{1}{\lambda}\log\mathbb{E}_P\left[\exp\left(\frac{\lambda}{m}\sum_{i=1}^{m}\varepsilon_i' f(z_i)\right)\right], \tag{27}$$

which implies

$$\sup_{Q\in\mathscr{Q}(\kappa)}\frac{1}{m}\sum_{i=1}^{m}\varepsilon_i'\mathbb{E}_Q f(z_i)\leq\frac{\kappa}{\lambda}+\frac{1}{\lambda}\log\mathbb{E}_P\left[\exp\left(\frac{\lambda}{m}\sum_{i=1}^{m}\varepsilon_i' f(z_i)\right)\right]. \tag{28}$$

Therefore, we have

$$\mathbb{P}_S\left(\sup_{Q\in\mathscr{Q}(\kappa)}\mathbb{E}_Q[\mathscr{L}_{\mathscr{D}}(f)-(1+c)\hat{\mathscr{L}}_S(f)]\geq t\right) \tag{29}$$

$$=\mathbb{P}_S\left(\sup_{Q\in\mathscr{Q}(\kappa)}\mathscr{L}_{\mathscr{D}}(\mathbb{E}_Q f)-(1+c)\hat{\mathscr{L}}_S(\mathbb{E}_Q f)\geq t\right) \tag{30}$$

$$\leq 4\mathbb{P}_{S,\epsilon}\left(\sup_{Q\in\mathscr{Q}(\kappa)}\frac{1}{m}\sum_{i=1}^{m}(\varepsilon_i-\frac{c'}{2+c'})\mathbb{E}_Q f(z_i)\geq\frac{t'}{2+c'}\right) \tag{31}$$

$$\leq 4\mathbb{P}_{S,\epsilon}\left(\frac{\kappa}{\lambda}+\frac{1}{\lambda}\log\mathbb{E}_P\left[\exp\left(\frac{\lambda}{m}\sum_{i=1}^{m}\varepsilon_i' f(z_i)\right)\right]\geq\frac{t'}{2+c'}\right) \tag{32}$$

$$=4\mathbb{P}_{S,\epsilon}\left(\log\mathbb{E}_P\left[\exp\left(\frac{\lambda}{m}\sum_{i=1}^{m}\varepsilon_i' f(z_i)\right)\right]\geq\frac{\lambda t'}{2+c'}-\kappa\right) \tag{33}$$

$$=4\mathbb{P}_{S,\epsilon}\left(\mathbb{E}_P\left[\exp\left(\frac{\lambda}{m}\sum_{i=1}^{m}\varepsilon_i' f(z_i)\right)\right]\geq\exp\left(\frac{\lambda t'}{2+c'}-\kappa\right)\right) \tag{34}$$

$$\underbrace{\leq}_{\text{Markov}} 4\exp\left(\kappa-\frac{\lambda t'}{2+c'}\right)\mathbb{E}_S\mathbb{E}_\epsilon\mathbb{E}_P\left[\exp\left(\frac{\lambda}{m}\sum_{i=1}^{m}\varepsilon_i' f(z_i)\right)\right]. \tag{35}$$

Now we use the result of Lemma A.2. For any $c_2$ such that $0<c_2<c$, if $t\geq\frac{1}{m}\frac{(1+c_2)^2}{c_2}$ and $\frac{c'}{c'+2}\geq\frac{\log\cosh(\lambda/m)}{\lambda/m}$ then we have

$$\mathbb{P}\left(\sup_{Q\in\mathscr{Q}(\kappa)}\mathbb{E}_Q[\mathscr{L}_{\mathscr{D}}(f)-(1+c)\hat{\mathscr{L}}_S(f)]\geq t\right)\leq 4\exp\left(\kappa-\frac{\lambda t'}{2+c'}\right), \tag{36}$$

where $c' = \frac{c-c_2}{1+c_2}$ and $t' = \frac{t}{2(1+c_2)}$. Now letting $4\exp\left(\kappa - \frac{\lambda t'}{2+c'}\right)$ equals to $\delta$, we have

$$t = 2(1+c_2)t' = 2(1+c_2)\left[\frac{2+c'}{\lambda}(\kappa + \log(4/\delta))\right]. \tag{37}$$

Now let $\frac{\lambda}{m} = C$, noting that

$$t \geq 2(1+c_2)\left[\frac{2+c'}{\lambda}\log(4/\delta)\right] \tag{38}$$

then we can choose $C$ small enough (clearly bounded away from 0) to satisfy

$$\frac{\log\cosh(C)}{C} \leq \frac{c'}{c'+2}, \quad C \leq \frac{2(1+c_2)(2+c')\log(4/\delta)}{(1+c_2)^2/c_2} \tag{39}$$

which guarantees both $t \geq \frac{1}{m}\frac{(1+c_2)^2}{c_2}$ and $\frac{c'}{c'+2} \geq \frac{\log\cosh(\lambda/m)}{\lambda/m}$.

Therefore, using such $C$ we have

$$\mathbb{P}\left(\sup_{Q \in \mathcal{Q}(\kappa)} \mathbb{E}_Q[\mathcal{L}_\mathcal{D}(f) - (1+c)\hat{\mathcal{L}}_S(f)] \geq \frac{2(1+c_2)(2+c')}{Cm}(\kappa + \log(4/\delta))\right) \leq \delta, \tag{40}$$

which implies

$$\sup_{Q \in \mathcal{Q}(\kappa)} \mathbb{E}_Q[\mathcal{L}_\mathcal{D}(f) - (1+c)\hat{\mathcal{L}}_S(f)] \leq \frac{C'}{m}(\kappa + \log(4/\delta)), \quad \text{w.p.} \quad 1-\delta. \tag{41}$$

where $C' = \frac{2(1+c_2)(2+c')}{C}$.

Finally, we combine all possible $\kappa$ using a union bound. Define $\Gamma_0 := \{Q : \text{KL}(Q||P) \leq 2\}$ and $\Gamma_j := \{Q : \text{KL}(Q||P) \in [2^j, 2^{j+1}]\}$ for $j \geq 1$. Let $\delta_j = 2^{-(j+1)}\delta$ so $\sum_{j=0}^\infty \delta_j = \delta$. Then for any $Q$ there is a $j_Q$ such that $Q \in \Gamma_{j_Q}$. Then by definition we have

$$\text{KL}(Q||P) \leq 2^{j_Q+1} \leq 2\max\{\text{KL}(Q||P), 1\} \tag{42}$$

$$\delta_{j_Q} = 2^{-(j_Q+1)}\delta \geq \frac{\delta}{2\max\{\text{KL}(Q||P), 1\}}. \tag{43}$$

Therefore, we have that for any $Q$, with probability $1-\delta$, we have

$$\mathcal{L}_\mathcal{D}(Q) - (1+c)\hat{\mathcal{L}}_S(Q) \leq \frac{C'}{m}(2\max\{\text{KL}(Q||P), 1\} + \log(8\max\{\text{KL}(Q||P), 1\}/\delta)) \tag{44}$$

$$\leq \frac{C'}{m}(2\max\{\text{KL}(Q||P), 1\} + \log(\max\{\text{KL}(Q||P), 1\}) + \log(8/\delta)). \tag{45}$$

Now we simplify the order without optimizing the constants, which gives

$$\log(\max\{\text{KL}(Q||P), 1\}) \leq \max\{\text{KL}(Q||P), 1\} \leq \text{KL}(Q||P) + 1. \tag{46}$$

Therefore, we have

$$\mathcal{L}_\mathcal{D}(f) - (1+c)\hat{\mathcal{L}}_S(f) \leq \frac{C_1}{m}\text{KL}(Q||P) + \frac{C_2}{m}\log(1/\delta) + \frac{C_3}{m}, \quad \text{w.p.} \quad 1-\delta. \tag{47}$$

**Lemma A.2.** *If* $k \geq \frac{\log\cosh(\lambda/m)}{\lambda/m}$*, then*

$$\mathbb{E}_P\mathbb{E}_S\mathbb{E}_\epsilon\left[\exp\left(\frac{\lambda}{m}\sum_{i=1}^m (\varepsilon_i - k)f(z_i)\right)\right] \leq 1. \tag{48}$$

*Proof.* Noting that $\{f(z_i)\}$ are independent Bernoulli random variables, we have

$$\mathbb{E}_P \mathbb{E}_S \mathbb{E}_\varepsilon \left[ \exp \left( \frac{\lambda}{m} \sum_{i=1}^{m} (\varepsilon_i - k) f(z_i) \right) \right] \tag{49}$$

$$\underbrace{=}_{\text{indep}} \mathbb{E}_P \prod_{i=1}^{m} \mathbb{E}_S \mathbb{E}_{\varepsilon_i} \left[ \exp \left( \frac{\lambda}{m} (\varepsilon_i - k) f(z_i) \right) \right] \tag{50}$$

$$\underbrace{=}_{\text{Rademacher}} \mathbb{E}_P \prod_{i=1}^{m} \mathbb{E}_S \left[ \frac{e^{(1-k)\frac{\lambda}{m} f(z_i)} + e^{-(1+k)\frac{\lambda}{m} f(z_i)}}{2} \right] \tag{51}$$

$$\underbrace{=}_{\text{Bernoulli}} \mathbb{E}_P \prod_{i=1}^{m} \left[ (1 - \mathbb{E}_S[f(z_i)]) + \left( \frac{e^{\frac{\lambda}{m}} + e^{-\frac{\lambda}{m}}}{2 e^{k\frac{\lambda}{m}}} \right) \mathbb{E}_S[f(z_i)] \right] \tag{52}$$

$$= \mathbb{E}_P \prod_{i=1}^{m} \left[ (1 - \mathbb{E}_S[f(z_i)]) + \frac{\cosh \left( \frac{\lambda}{m} \right)}{e^{k\frac{\lambda}{m}}} \mathbb{E}_S[f(z_i)] \right], \tag{53}$$

which is upper bounded by 1 if we choose $k$ such that

$$\cosh(\lambda/m) = \frac{e^{\lambda/m} + e^{-\lambda/m}}{2} \leq e^{k\lambda/m}. \tag{54}$$

That is

$$k \geq \frac{\log \cosh(\lambda/m)}{\lambda/m}. \tag{55}$$

$\square$

## A.2 Proof of Theorem 4.3

We first present some lemmas that will be used in the later proof.

**Lemma A.3** (Shifted-Flatness Inequality). *Consider a function $f : \mathscr{Z} \to [0,1]$, constants $h \in [0,1]$ and $c_2 > 0$, if $t \geq \frac{(1+c_2)(1+c_2 h^2)}{m c_2 h^2}$, we have*

$$\mathbb{P}_S \left( \mathscr{L}_{\mathscr{D}}(f) - (1+c_2) \hat{\mathscr{L}}_S(f) + c_2(1-h^2) \hat{\mathscr{L}}_S(f^2) \geq \frac{t}{2} \right) \leq \frac{1}{2}. \tag{56}$$

*Proof.* Let $v = c_2 \mathscr{L}_{\mathscr{D}}(f) - c_2(1-h^2) \mathscr{L}_{\mathscr{D}}(f^2) = c_2 \mathscr{L}_{\mathscr{D}}(f - (1-h^2)f^2)$. Then, we have

$$\mathbb{P} \left( \mathscr{L}_{\mathscr{D}}(f) - (1+c_2) \hat{\mathscr{L}}_S(f) + c_2(1-h^2) \hat{\mathscr{L}}_S(f^2) \geq \frac{t}{2} \right) \tag{57}$$

$$= \mathbb{P} \left( \mathscr{L}_{\mathscr{D}}(f) - \frac{c_2(1-h^2)}{1+c_2} \mathscr{L}_{\mathscr{D}}(f^2) - \hat{\mathscr{L}}_S(f) + \frac{c_2(1-h^2)}{1+c_2} \hat{\mathscr{L}}_S(f^2) \geq \frac{t/2+v}{1+c_2} \right) \tag{58}$$

$$= \mathbb{P} \left( \mathbb{E}_{z \sim \mathscr{D}}(f(z) - \frac{c_2(1-h^2)}{1+c_2} f(z)^2) - \frac{1}{m} \sum_{i=1}^{m} (f(z_i) - \frac{c_2(1-h^2)}{1+c_2} f(z_i)^2) \geq \frac{t/2+v}{1+c_2} \right). \tag{59}$$

Because $f(z_i) - \frac{c_2(1-h^2)}{1+c_2} f(z_i)^2, i = 1, \ldots, m$ are i.i.d. random samples, using Chebyshev's inequality together with $4ab \leq (a+b)^2$ and $f \in [0,1]$, the formula above is upper bounded by

$$\frac{(1+c_2)^2 \text{Var}(f - \frac{c_2(1-h^2)}{1+c_2} f^2)}{m(t/2+v)^2} \leq \frac{(1+c_2)^2 \mathscr{L}_{\mathscr{D}}(f - \frac{c_2(1-h^2)}{1+c_2} f^2)^2}{2mvt} \leq \frac{(1+c_2)^2 \mathscr{L}_{\mathscr{D}}(f - \frac{c_2(1-h^2)}{1+c_2} f^2)}{2mvt}. \tag{60}$$

We can further decompose the term in the numerator by

$$\mathscr{L}_{\mathscr{D}}(f - \frac{c_2(1-h^2)}{1+c_2} f^2) = \frac{c_2}{1+c_2} \mathscr{L}_{\mathscr{D}}(f - (1-h^2)f^2) + \frac{1}{1+c_2} \mathscr{L}_{\mathscr{D}}(f) \tag{61}$$

$$\leq \frac{1}{1+c_2} v + \frac{1}{1+c_2} \frac{1}{c_2 h^2} v = \frac{c_2 h^2 + 1}{(1+c_2) c_2 h^2} v, \tag{62}$$

Therefore the lemma follows directly from $t \geq \frac{(1+c_2)(1+c_2 h^2)}{m c_2 h^2}$. $\square$

**Lemma A.4** (New Shifted Symmetrization in Deviation). *Fix constants $c > c_2 > 0$, $h \in [0,1]$, if $t \geq \frac{(1+c_2)(1+c_2 h^2)}{mc_2 h^2}$, we have*

$$\frac{1}{4}\mathbb{P}_S \left( \sup_{f \in \mathscr{F}} \mathscr{L}_{\mathscr{D}}(f) - (1+c)\hat{\mathscr{L}}_S(f) + c(1-h^2)\hat{\mathscr{L}}_S(f^2) \geq t \right)$$

$$\leq \mathbb{P}_{S,\epsilon} \left( \sup_{f \in \mathscr{F}} \left[ \frac{1}{m}\sum_{i=1}^m \varepsilon_i \left( (1+c')f(z_i) - c'(1-h^2)f^2(z_i) \right) - c''\hat{\mathscr{L}}_S \left( f - (1-h^2)f^2 \right) \right] \geq \frac{t}{4} \right), \quad (63)$$

*where $c' = \frac{c+c_2}{2}, c'' = \frac{c-c_2}{2}$, $\epsilon := \{\varepsilon_i\}_{i=1}^m$, in which $\{\varepsilon_i\}$ are independent Rademacher random variables.*

*Proof.* Consider a random set $S' = \{z_i'\}_{i=1}^m \in \mathscr{D}^m$, in Lemma A.3 we have shown that

$$\mathbb{P}_{S'} \left( \mathscr{L}_{\mathscr{D}}(f) - (1+c_2)\hat{\mathscr{L}}_{S'}(f) + c_2(1-h^2)\hat{\mathscr{L}}_{S'}(f^2) \geq \frac{t}{2} \right) \leq \frac{1}{2}. \quad (64)$$

Therefore, we can get

$$\frac{1}{4}\mathbb{P}_S \left( \sup_{f \in \mathscr{F}} \mathscr{L}_{\mathscr{D}}(f) - (1+c)\hat{\mathscr{L}}_S(f) + c(1-h^2)\hat{\mathscr{L}}_S(f^2) \geq t \right) \quad (65)$$

$$\leq \frac{1}{2}\mathbb{P}_{S,S'} \left( \sup_{f \in \mathscr{F}} (1+c_2)\hat{\mathscr{L}}_{S'}(f) - c_2(1-h^2)\hat{\mathscr{L}}_{S'}(f^2) - (1+c)\hat{\mathscr{L}}_S(f) + c(1-h^2)\hat{\mathscr{L}}_S(f^2) \geq \frac{t}{2} \right) \quad (66)$$

$$= \frac{1}{2}\mathbb{P}_{S,S'} \left[ \sup_{f \in \mathscr{F}} (1 + \frac{c+c_2}{2}) \left( \hat{\mathscr{L}}_S(f) - \hat{\mathscr{L}}_{S'}(f) \right) - \frac{c+c_2}{2}(1-h^2) \left( \hat{\mathscr{L}}_S(f^2) - \hat{\mathscr{L}}_{S'}(f^2) \right) \right. \quad (67)$$

$$\left. - \frac{c-c_2}{2}\hat{\mathscr{L}}_{S'} \left( f - (1-h^2)f^2 \right) - \frac{c-c_2}{2}\hat{\mathscr{L}}_S \left( f - (1-h^2)f^2 \right) \geq \frac{t}{2} \right] \quad (68)$$

$$\leq \mathbb{P}_{S,\epsilon} \left( \sup_{f \in \mathscr{F}} \left[ \frac{1}{m}\sum_{i=1}^m \varepsilon_i \left( (1+c')f(z_i) - c'(1-h^2)f^2(z_i) \right) - c''\hat{\mathscr{L}}_S \left( f - (1-h^2)f^2 \right) \right] \geq \frac{t}{4} \right), \quad (69)$$

*where $c' = \frac{c+c_2}{2}, c'' = \frac{c-c_2}{2}$, and the last inequality is by the symmetrization argument.* $\square$

**Lemma A.5.** *For constants $h \in (0,1], h^2 c > c_2 > 0$, let $C = \frac{h^2 c - c_2}{2(1+h^2 c)(1+c_2)}$, if $0 < \frac{\lambda}{m} < C$, then given independent Bernoulli random variables $X_i$, for any random variables $Y_i \in [0,1]$,*

$$\mathbb{E}_\varepsilon \mathbb{E}_X \mathbb{E}_{Y|X,\varepsilon} \left[ \exp \left( \frac{\lambda}{m}\sum_{i=1}^m X_i \left[ (\varepsilon_i + \varepsilon_i'') - \varepsilon_i''(1-h^2)Y_i \right] \right) \right] \leq 1, \quad (70)$$

*where $\{\varepsilon_i\}$ are i.i.d. Rademacher random variables and $\varepsilon_i'' = \varepsilon_i \frac{c+c_2}{2} - \frac{c-c_2}{2}$.*

*Proof.* Note when $X_i = 0$, the value of $Y_i$ has no effect onto LHS. When $X_i = 1$,

$$(\varepsilon_i + \varepsilon_i'') - \varepsilon_i''(1-h^2)Y_i = (1+c_2) - c_2(1-h^2)Y_i, \text{ if } \varepsilon_i = 1, \quad (71)$$

$$(\varepsilon_i + \varepsilon_i'') - \varepsilon_i''(1-h^2)Y_i = -(1+c) + c(1-h^2)Y_i, \text{ if } \varepsilon_i = -1, \quad (72)$$

Therefore, we have

$$(\varepsilon_i + \varepsilon_i'') - \varepsilon_i''(1-h^2)Y_i \leq (\varepsilon_i + \varepsilon_i'') - \varepsilon_i''(1-h^2)\frac{1-\varepsilon_i}{2}. \quad (73)$$

Denoting $\mu_i = \mathbb{E}[X_i]$, by the monotonicity of the exponential function, we have

$$\mathbb{E}_\varepsilon \mathbb{E}_X \mathbb{E}_{Y|X,\varepsilon} \left[ \exp\left( \frac{\lambda}{m} \sum_{i=1}^m X_i \left[ (\varepsilon_i + \varepsilon_i'') - \varepsilon_i'' (1-h^2) Y_i \right] \right) \right] \tag{74}$$

$$\leq \mathbb{E}_\varepsilon \mathbb{E}_X \left[ \exp\left( \frac{\lambda}{m} \sum_{i=1}^m X_i \left[ (\varepsilon_i + \varepsilon_i'') - \varepsilon_i'' (1-h^2) \frac{1-\varepsilon_i}{2} \right] \right) \right] \tag{75}$$

$$= \prod_{i=1}^m \left[ 1 - \mu_i + \frac{\mu_i}{2} \exp\left( \frac{\lambda}{m}(1+c_2) \right) + \frac{\mu_i}{2} \exp\left( -\frac{\lambda}{m}(1+h^2 c) \right) \right], \tag{76}$$

For the formula upper bounded by 1, it is sufficient to prove

$$\exp\left( \frac{\lambda}{m}(1+c_2) \right) + \exp\left( -\frac{\lambda}{m}(1+h^2 c) \right) \leq 2. \tag{77}$$

Because we have $e^x \geq x+1$, thus $e^{-x} \leq \frac{1}{1+x}$ for $x > -1$ and $e^x \leq \frac{1}{1-x}$ for $x < 1$. Therefore, it is sufficient to have

$$\frac{1}{1 - \frac{\lambda}{m}(1+c_2)} + \frac{1}{1 + \frac{\lambda}{m}(1+h^2 c)} \leq 2, \quad \frac{\lambda}{m}(1+c_2) \leq 1. \tag{78}$$

Thus, we know the argument holds for

$$\frac{\lambda}{m} \leq \min\left( \frac{h^2 c - c_2}{2(1+h^2 c)(1+c_2)}, \frac{1}{1+c_2} \right) = \frac{h^2 c - c_2}{2(1+h^2 c)(1+c_2)}, \tag{79}$$

where the last equality holds when $c_2 < h^2 c$. $\qquad\square$

Now we are ready for the proof of Theorem 4.3. Denoting $g(\cdot) = \mathbb{E}_Q f(\cdot)$, one can write

$$\mathscr{L}_{\mathscr{D}}(Q) - \hat{\mathscr{L}}_S(Q) - \frac{c}{m} \sum_{i=1}^m \mathbb{E}_Q [f(z_i) - (1+h)\mathbb{E}_Q f(z_i)]^2 \tag{80}$$

$$= \mathscr{L}_{\mathscr{D}}(Q) - (1+c)\hat{\mathscr{L}}_S(Q) + \frac{c(1-h^2)}{m} \sum_{i=1}^m (\mathbb{E}_Q f(z_i))^2 \tag{81}$$

$$= \mathscr{L}_{\mathscr{D}}(g) - (1+c)\hat{\mathscr{L}}_S(g) + \frac{c(1-h^2)}{m} \sum_{i=1}^m (g(z_i))^2 \tag{82}$$

$$= \mathscr{L}_{\mathscr{D}}(g) - (1+c)\hat{\mathscr{L}}_S(g) + c(1-h^2)\hat{\mathscr{L}}_S(g^2). \tag{83}$$

Recall that for $\kappa > 0$, we have defined $\mathscr{Q}(\kappa) = \{Q : \mathrm{KL}(Q||P) \leq \kappa\}$. We start from the formula,

$$\mathbb{P}_S \left[ \sup_{Q \in \mathscr{Q}(\kappa)} \mathscr{L}_{\mathscr{D}}(g) - (1+c)\hat{\mathscr{L}}_S(g) + c(1-h^2)\hat{\mathscr{L}}_S(g^2) \geq t \right]. \tag{84}$$

By Lemma A.4, let $c' = \frac{c+c_2}{2}, c'' = \frac{c-c_2}{2}$, and $\{\varepsilon_i\}$ being i.i.d. Rademacher random variables, we have

$$\mathbb{P}_S \left[ \sup_{Q \in \mathscr{Q}(\kappa)} \mathscr{L}_{\mathscr{D}}(g) - (1+c)\hat{\mathscr{L}}_S(g) + c(1-h^2)\hat{\mathscr{L}}_S(g^2) \geq t \right] \tag{85}$$

$$\leq 4\mathbb{P}_{S,\varepsilon} \left[ \sup_{Q \in \mathscr{Q}(\kappa)} \frac{1}{m} \sum_{i=1}^m \varepsilon_i \left( (1+c') g(z_i) - c'(1-h^2)g(z_i)^2 \right) - c''\hat{\mathscr{L}}_S(g - (1-h^2)g^2) \geq \frac{t}{4} \right]. \tag{86}$$

Plugging into $g = \mathbb{E}_Q f(\cdot)$ yields

$$\sup_{Q \in \mathscr{Q}(\kappa)} \frac{1}{m} \sum_{i=1}^m \varepsilon_i \left( (1+c') g(z_i) - c'(1-h^2)g(z_i)^2 \right) - c''\hat{\mathscr{L}}_S(g - (1-h^2)g^2) \tag{87}$$

$$= \sup_{Q \in \mathscr{Q}(\kappa)} \frac{1}{m} \sum_{i=1}^m [\varepsilon_i(1+c') - c''] \mathbb{E}_Q f(z_i) - [\varepsilon_i c' - c''](1-h^2)[\mathbb{E}_Q f(z_i)]^2. \tag{88}$$

Given $\epsilon = \{\varepsilon_i\}_{i=1}^m, \mathbf{z} = \{z_i\}_{i=1}^m$, we suppose $\hat{Q}(\epsilon, \mathbf{z})$ achieves the supremum above (if the supremum cannot be achieved, one can use a sequence of $\{\hat{Q}_i(\epsilon, \mathbf{z})\}$ to approximate arbitrarily close to the supremum). Using

$$\varepsilon_i'' := \varepsilon_i c' - c'' = \varepsilon_i \frac{c_1 + c_2}{2} - \frac{c_1 - c_2}{2}, \tag{89}$$

we have

$$\sup_{Q \in \mathscr{Q}(\kappa)} \frac{1}{m} \sum_{i=1}^m [\varepsilon_i (1 + c') - c''] \mathbb{E}_Q f(z_i) - [\varepsilon_i c' - c''] (1 - h^2) [\mathbb{E}_Q f(z_i)]^2 \tag{90}$$

$$\leq \sup_{Q \in \mathscr{Q}(\kappa)} \frac{1}{m} \sum_{i=1}^m [\varepsilon_i (1 + c') - c''] \mathbb{E}_Q f(z_i) - [\varepsilon_i c' - c''] (1 - h^2) \mathbb{E}_Q f(z_i) \mathbb{E}_{\hat{Q}(\epsilon, \mathbf{z})} f(z_i) \tag{91}$$

$$= \sup_{Q \in \mathscr{Q}(\kappa)} \mathbb{E}_Q \left[ \frac{1}{m} \sum_{i=1}^m (\varepsilon_i + \varepsilon_i'') f(z_i) - \varepsilon_i'' (1 - h^2) f(z_i) \mathbb{E}_{\hat{Q}(\epsilon, \mathbf{z})} f(z_i) \right] \tag{92}$$

$$\leq \frac{\kappa}{\lambda} + \frac{1}{\lambda} \log \mathbb{E}_P \left[ \exp \left( \frac{\lambda}{m} \sum_{i=1}^m f(z_i) \left[ (\varepsilon_i + \varepsilon_i'') - \varepsilon_i'' (1 - h^2) \mathbb{E}_{\hat{Q}(\epsilon, \mathbf{z})} f(z_i) \right] \right) \right], \tag{93}$$

where the last inequality follows from duality for convex optimization [7, Chp. 5].

Therefore, we have

$$\mathbb{P} \left[ \sup_{Q \in \mathscr{Q}(\kappa)} \mathscr{L}_\mathscr{D}(Q) - (1 + c) \hat{\mathscr{L}}_S(Q) + \frac{c(1 - h^2)}{m} \sum_{i=1}^m (\mathbb{E}_Q f(z_i))^2 \geq t \right] \tag{94}$$

$$\leq 4 \mathbb{P} \left[ \frac{\kappa}{\lambda} + \frac{1}{\lambda} \log \mathbb{E}_P \left[ \exp \left( \frac{\lambda}{m} \sum_{i=1}^m f(z_i) \left[ (\varepsilon_i + \varepsilon_i'') - \varepsilon_i'' (1 - h^2) \mathbb{E}_{\hat{Q}(\epsilon, \mathbf{z})} f(z_i) \right] \right) \right] \geq \frac{t}{4} \right] \tag{95}$$

$$\leq 4 \exp \left( \kappa - \frac{\lambda t}{4} \right) \mathbb{E}_S \mathbb{E}_\epsilon \mathbb{E}_P \left[ \exp \left( \frac{\lambda}{m} \sum_{i=1}^m f(z_i) \left[ (\varepsilon_i + \varepsilon_i'') - \varepsilon_i'' (1 - h^2) \mathbb{E}_{\hat{Q}(\epsilon, \mathbf{z})} f(z_i) \right] \right) \right] \tag{96}$$

$$\leq 4 \exp \left( \kappa - \frac{\lambda t}{4} \right), \tag{97}$$

where the last inequality comes from Lemma A.5 by considering $X_i$ as $f(z_i)$ and $Y_i$ as $\mathbb{E}_{\hat{Q}(\epsilon, \mathbf{z})} f(z_i)$, with

$$C := \frac{\lambda}{m} \leq \frac{h^2 c - c_2}{2(1 + h^2 c)(1 + c_2)}. \tag{98}$$

Now let $4 \exp \left( \kappa - \frac{\lambda t}{4} \right)$ equals to $\delta$, we have

$$t = \frac{4}{\lambda} \left( \kappa + \log \frac{4}{\delta} \right) = \frac{4}{Cm} \left( \kappa + \log \frac{4}{\delta} \right). \tag{99}$$

Note that the shifted symmetrization inequality requires $t \geq \frac{(1 + c_2)(1 + c_2 h^2)}{mc_2 h^2}$ by Lemma A.3. Combining with the previous requirement for $C$ together, we have

$$C \leq \min \left( \frac{h^2 c - c_2}{2(1 + h^2 c)(1 + c_2)}, \frac{4 c_2 h^2}{(1 + c_2)(1 + c_2 h^2)} \left( \kappa + \log \frac{4}{\delta} \right) \right). \tag{100}$$

Using such $C$ we have with probability at least $1 - \delta$,

$$\sup_{Q \in \mathscr{Q}(\kappa)} \mathbb{E}_Q \mathscr{L}_\mathscr{D}(f) - \mathbb{E}_Q \hat{\mathscr{L}}_S(f) - \frac{c}{m} \sum_{i=1}^m \mathbb{E}_Q [f(z_i) - (1 + h) \mathbb{E}_Q f(z_i)]^2 \leq \frac{4}{Cm} \left( \kappa + \log \frac{4}{\delta} \right). \tag{101}$$

Finally we combine all possible $\kappa$ using a union bound. Define $\Gamma_0 = \{Q : \mathrm{KL}(Q \| P) \leq 2\}$ and $\Gamma_j = \{Q : \mathrm{KL}(Q \| P) \in [2^j, 2^{j+1}]\}$ for $j \geq 1$. Let $\delta_j = 2^{-(j+1)} \delta$ so that $\sum_{j=0}^\infty \delta_j = \delta$. Then for any $Q$

there is a $j_Q$ such that $Q \in \Gamma_{j_Q}$. Then we have

$$\mathrm{KL}\left(Q||P\right) \leq 2^{j_Q+1} \leq 2\max(\mathrm{KL}\left(Q||P\right),1) \tag{102}$$

$$\delta_{j_Q} = 2^{-(j_Q+1)}\delta \geq \frac{\delta}{2\max(\mathrm{KL}\left(Q||P\right),1)}, \tag{103}$$

Therefore with probability at least $1 - \delta$ over draws of $S$, for any $Q$,

$$\mathbb{E}_Q \mathscr{L}_{\mathscr{D}}(f) \leq \mathbb{E}_Q \hat{\mathscr{L}}_S(f) + \frac{c}{m}\sum_{i=1}^{m} \mathbb{E}_Q[f(z_i) - (1+h)\mathbb{E}_Q f(z_i)]^2$$

$$+ \frac{4}{Cm}\left[2\max(\mathrm{KL}\left(Q||P\right),1) + \log \frac{8\max(\mathrm{KL}\left(Q||P\right),1)}{\delta}\right] \tag{104}$$

$$\leq \mathbb{E}_Q \hat{\mathscr{L}}_S(f) + \frac{c}{m}\sum_{i=1}^{m} \mathbb{E}_Q[f(z_i) - (1+h)\mathbb{E}_Q f(z_i)]^2 + \frac{4}{Cm}\left[3\mathrm{KL}\left(Q||P\right) + \log \frac{1}{\delta} + 5\right], \tag{105}$$

provided that

$$C \leq \min(\frac{h^2 c - c_2}{2(1+h^2 c)(1+c_2)}, \frac{4c_2 h^2}{(1+c_2)(1+c_2 h^2)}(\kappa + \log \frac{4}{\delta})). \tag{106}$$

Therefore, it is sufficient if

$$C = \frac{2h^4 c}{1+16h^2 c}, \quad c_2 = \frac{h^2 c}{1+16h^2 c}. \tag{107}$$