[Reviews · NeurIPS 2019]

Reviewer 1



This paper is clearly original since it provides a new proof of a well-known result (Catoni's bound, recovered up to constants), a new definition of flatness (h-flatness) of the landscape where the posterior concentrates and a new PAC-Bayes bound associated to the notion of flatness. This paper is a nice theoretical contribution to the PAC-Bayesian theory of learning. The details of the proofs, that are technically sound (use of symmetrization for deviation inequalities, theory of shifted empirical process, duality formula for the kullback-Leibler divergence, peeling with respect the values of the KL divergence), are deferred to the Appendix and condensed arguments of proofs are written in the main part. However, the comparison of the new bound (Theorem 4.3) with Catoni's bound remains at rather heuristic level. Indeed, it is proved that the new bound is not worst than Catoni's bound, but it is only assumed and not clearly demonstate that the new bound can actually be stronger than Catoni's bound in favorable cases. I have also a question in mind about the scheme of proof exposed in Section 3. To bypass the suboptimality of the bounded difference inequality, the paper advocates to use control by Rademacher quantities directly in the deviation inequalities. But what about Talagrand's type concentration inequalities for the supremum of the empirical process, that would allow (classically through a peeling argument) to take into account the localization of the variance of the classes (KL neighborhoods). This is the classical way to produce fast rates from empirical process theory, so I wonder if some adaptation could be worked out here in the PAC-Bayes framework? This paper is essentially well written and well organized. The notations are carfully introduced and many references to existing, connected works are provided. However, I have three remarks concerning the presentation, that are rather minor. Constant c does not appear in display (11) (that refers to Lemma A.2 in the supplementary file), on contrary to what is anounced (l. 158: "as long as c>0"). This should be fixed. Also, l. 175-176, the sentence that begins with "For fixed $h>0$ [...]" is unclear to me and should be rewritten. Finally, I think that Display (25) could help the reader to understand the notion of flatness if it would be placed right after Remark 4.2. I think that the proof of Proposition 3.1, the new bound of Theorem 4.3 and the definition of flatness provided in Definition 4.1 are interesting and important theoretical contributions, successfully investigating the links between two major lines of research, Rademacher complexities and PAC-Bayes bounds.

Reviewer 2



The paper investigates the connection between PAC-Bayes and Rademacher complexities, two framework in statistical learning theory to upper bound the generalization error of predictors. The paper is quite pleasant to read, and clarity is remarkable. Detailed comments: * The paper investigates the binary classification case, with the loss taking values in $\{0,1\}$. Is there hope to transpose the present results to more general losses? It is unclear whether the technique used by the authors depends on the fact that the loss takes only two values -- or is bounded. There are also a few papers on PAC-Bayes with unbounded losses, see e.g. Alquier and Guedj (2018), "Simpler PAC-Bayesian bounds for hostile data", Machine Learning and references therein (note: slow rates). * Eq. (9) is reminiscent of the Legendre transform of the Kullback-Leibler divergence which has been long known and used in the PAC-Bayes literature. See for example Lemma 1 in the recent survey "A primer on PAC-Bayesian learning" (Guedj, 2019), https://arxiv.org/abs/1901.05353 Is there a connection here? * Lines 182-186: I disagree with that comment. Even if $\frac{c}{m}\sum_{i=1}^m \mathbb{E}_Q[f(z_i)-(1+h)\mathbb{E}_Q f(z_i)]^2$ is negligible, the right-hand side in (14) is still larger than the corresponding term in Catoni's bound, due to numerical constants (in the present paper: $\frac{4}{C}(3\mathbf{KL}(Q||P) + \log(1/\delta)) + 5$ compared to $(1+t)(\mathbf{KL}(Q||P)+\log(1/\deta))$ in Catoni's bound, where $t$ is arbitrarily close to 0). * Related work: a few PAC-Bayes references seem to be missing (see the afomentioned survey Guedj, 2019 and references therein). In particular, Tolstikhin and Seldin (2013), "PAC-Bayesian-empirical-Bernstein inequality", NIPS (PAC-Bayes bound with a variance term) seems relevant as the authors prove a PAC-Bayes bound with a variance term. I am unsure why the authors of the present paper insist that their notion of "flatness" is "\emph{not} related to the variance of the randomized classifier". While it is technically true, the terms are still quite similar and one would expect the actual values to be close when computing the bounds (especially when $h$ is small). The paragraph on lines 249-262 could be enriched with additional comments and comparisons. In particular, see Theorem 4 in Tolstikhin and Seldin (2013). * References: please remove "et al." for [3] and [39]. === after rebuttal === I thank the authors for taking the time to address my remarks in their rebuttal. After reading other reviews, rebuttal and engaging in discussion, my score remains unchanged. Nevertheless, I must say I am disappointed by some parts of the rebuttal. Point (5): I don't agree with the rebuttal and I think discarding a comparison with Tolstikhin and Seldin's bound on the ground that their term may be large whereas the authors' is zero is wrong: I see no evidence for that claim. Last but not least, see Alquier and Guedj (2018), propositions 1 and 2: PAC-Bayes bounds *can* be easily specified to the ERM case, contrary to what is written in the rebuttal. I still have an overall positive opinion on the paper but I sure would like to see a more thorough and rigorous comparison with existing bounds, both theoretically and numerically.

Reviewer 3



Although the obtained PAC-Bayesian bound is not novel, the new proof technique can enrich the comprehension both PAC-Bayes and Rademacher generalization theories and foster the emergence of new ideas. The authors made a substantial effort to expose their results in a pedagogical manner. I appreciate the choice to write proofs outlines in the main paper, and the full proofs in the appendix. I did not check the proofs in detail, but it appears to be a rigorous piece of work. That being said, the paper would be richer if the results were discussed more. The comparison to other work could be expended. In particular, here are some points that I would like to be discussed: - How the current work relates to Grunwald and Mehta "unified PAC-Bayesian–Rademacher–Shtarkov–MDL" approach, cited as [16] within the paper (but not discussed)? - How the notion of flatness relates to existing ones in the deep learning and/or optimization literature? It is said that the result is "inspired" by Dzugaite and Roy (2017), but it is unclear if the connection is solely thematic. - Why the proposed approach is more appropriate than "classical" ones to obtain the "flatness" result? Minor comments: - Lines 80-85: When discussing the convergence rate of Catoni's bound, it should be mentioned that the constant C must be small in order for the bound to converge (e.g. as mentioned in [26]). - Equation 4: For the reader benefit, I suggest clarifying that the z_i are the elements of S from the expectation according to the data distribution. - Lines 127-128: It would be more accurate to say that c / (1-e^-c) <= 1+c instead of saying that writing both are equivalent. - There are some inaccuracies the references: Ref [7]: Published. See https://projecteuclid.org/euclid.lnms/1199996410 Ref [16]: Published. See http://proceedings.mlr.press/v98/grunwald19a.html Ref [19]: The publication year is 2008. -------------- I am satisfied with the author’s rebuttal. As they did commit to enrich the paper, by answering my comments and those of my two fellow reviewers, I raised score to from 6 to 7. About the point (7) of the rebuttal: It is disappointing that the authors sell their flatness result by suggesting a connection with Dziugaite and Roy (2017), but it turns out that the connection still needs to be made rigorously. That being said, I appreciate the authors honesty concerning their preliminary unsuccessful experiments. I agree that this is beyond the scope of the paper and can be regarded as future work. However, the author should revise slightly the way they present their result, and be clear that it is solely remotely connected to Dziugaite and Roy (2017). About the point (8) of the rebuttal: I agree with the authors that my question relating to the "classical approaches" was lacking clarity. More than raising an "open problem", I suggest to comment on which technical aspect of their Rademacher approach allows to derive the flatness result is not (yet) possible by a direct PAC-Bayes approach. It would help the readers to appreciate the result.

[Author Response · NeurIPS 2019]

1) Comparison with Catoni's Bound (R1 & R2 & R3): Catoni's bound has the form $(1 + c_c)\hat{\mathcal{L}}_S Q + \frac{\mathcal{C}_c}{m}(\text{KL}(Q\|P) + \log\frac{1}{\delta})$, while our bound (Eq. (25)) can be written $(1 + c_r)\hat{\mathcal{L}}_S Q - \frac{c_r(1-h^2)}{m}\sum_{i=1}^m(\mathbb{E}_Q f(z_i))^2 + \frac{\mathcal{C}_r}{m}(\text{KL}(Q\|P) + \log\frac{1}{\delta} + 1)$. Here $c_c, c_r$ inflate the empirical risk and $\mathcal{C}_c, \mathcal{C}_r$ are constants. Let $T_m$ be $\frac{c_r(1-h^2)}{m}\sum_{i=1}^m(\mathbb{E}_Q f(z_i))^2$. Note that $c_c$ and $c_r$ must be fixed before seeing the data. Assuming we equate the inflation of the empirical risk, i.e., $c_c = c_r$, the proposed bound is tighter than Catoni's bound provided $m > \frac{1}{T_m}\left((\mathcal{C}_r - \mathcal{C}_c)\left(\text{KL}(Q\|P) + \log\frac{1}{\delta}\right) + \mathcal{C}_r\right)$. If $T_m$ converges to a positive number (a reasonable assumption), then our proposed bound will be tighter for sufficiently many samples. If we assume $c_c \neq c_r$, our bound can still be tighter than Catoni's bound under more involved conditions.

2) Talagrand-type concentration inequalities (R1): We thank the reviewer for this valuable comment. As the reviewer anticipates, some adaptation of Talagrand's result seems to be necessary. We can raise this as an open question.

3) From binary loss to bounded or unbounded losses (R2): Our main results for binary loss can be extended to $[0, 1]$-valued (i.e., bounded) loss, but with a different constant $C = \frac{4h^2 c}{9(1+h^2 c)(1+h^2 c/9)}$ that has the same interpretation as in Eq. (14). Briefly, the proof relies on Jensen's inequality $\mathcal{L}_\mathcal{D}(Q) - \hat{\mathcal{L}}_S(Q) - \frac{c}{m}\sum_{i=1}^m \mathbb{E}_Q[f(z_i) - (1+h)\mathbb{E}_Q f(z_i)]^2 \leq \mathcal{L}_\mathcal{D}(Q) - \hat{\mathcal{L}}_S(Q) - \frac{ch^2}{m}\sum_{i=1}^m(\mathbb{E}_Q f(z_i))^2$. Then, by a symmetrization technique similar to that in Eq. (17), we get

$$\frac{1}{4}\mathbb{P}_S\left(\sup_{g\in\mathcal{G}_\kappa}\mathcal{L}_\mathcal{D}(g) - \hat{\mathcal{L}}_S(g) - ch^2\hat{\mathcal{L}}_S(g^2) \geq t\right) \leq \mathbb{P}_{S,\epsilon}\left(\sup_{g\in\mathcal{G}_\kappa}\left[\frac{1}{m}\sum_{i=1}^m \epsilon_i\left(g(z_i) + c'h^2 g^2(z_i)\right) - c''h^2\hat{\mathcal{L}}_S\left(g^2\right)\right] \geq \frac{t}{4}\right)$$

where $c' = \frac{c+c_2}{2}, c'' = \frac{c-c_2}{2}$. This inequality enables the use of KL's Legendre transform, as in Eq. (23). The bound for $[0, 1]$-valued loss can then be derived following similar techniques in the paper. We're happy to include these details or leave them out as the reviewers see fit. For unbounded losses, it might be possible to extend our results under a sub-Gaussian assumption, but we prefer not to speculate. In the revised paper, we discuss extensions to general loss functions and cite Alquier and Guedj (2018).

4) Connections between Eq. (9) and Lemma 1 of (Guedj, 2019) (R2): Thanks for pointing out this connection. We now cite Guedj (2019) in our revision.

5) Comparison with Tolstikhin and Seldin, 2013 (R2): Thank you for pointing out this missing reference. Note that the empirical variance (as appears in Thm. 4 of Tolstikhin and Seldin, 2013) and our "flatness" are distinct. The former is $\mathbb{E}_Q \frac{1}{m}\sum_{i=1}^m[f(z_i) - \frac{1}{m}\sum_{i=1}^m f(z_i)]^2$, while our "flatness" is $\frac{1}{m}\sum_{i=1}^m \mathbb{E}_Q[f(z_i) - \mathbb{E}_Q f(z_i)]^2$. It is possible for the second quantity ("flatness") to be zero, even when the first quantity is large. We now cite Tolstikhin and Seldin (2013) and highlight the relationship.

6) Connections to Grunwald and Mehta, 2019 (R3): Grunwald and Mehta propose a novel notion of complexity in terms of a "luckiness function", which generalizes the "prior" in PAC-Bayes and unifies the classical Rademacher complexity bound for ERM into the same framework. On the other hand, the paper is not directly related to deriving PAC-Bayes bounds using Rademacher-process approaches, hence not comparable to our work. However, it is certainly of great interest to study if our PAC-Bayes work can be extended to their more general framework in terms of "luckiness functions". We will add the discussions in the revised paper.

7) Connections to Dziugaite and Roy, 2017 (R2 & R3): We modified the code of Dziugaite and Roy (2017) and determined that the posterior they find is not "$h$-flat" in our sense. After some investigation, we believe the reason is that they are optimizing a PAC-Bayes bound and due to the poor prior choice, they underfit, and as a result, the posterior they find corresponds to a Gaussian with large variance for many parameters that are essentially "useless". We think investigating this and other empirical questions further is an interesting and open avenue of research, though well beyond the scope of this paper.

8) How is the proposed approach "more appropriate than "classical" approaches"? (R3): We're not entirely clear on the question, but here is our best attempt. We're happy to add further discussion if the reviewer can expand their question in their update. We cannot at present derive our "flatness" bound by a direct PAC-Bayes approach, without going through the Rademacher argument. We now raise this as an open problem.

9) Choosing a Dirac mass as a posterior would enable comparison with ERM (R2): In order for there to exist a (data-independent) prior $P$ that, with high probability, dominates a Dirac mass concentrated on a random point $\eta$ (thus yielding a finite KL divergence term), $\eta$ must lie in a countable set with high probability. In general, ERMs do not satisfy this property. In order to study ERM using PAC-Bayes bounds, one usually relates the risk and empirical risk of a Gibbs classifier to the ERM. Standard approaches exploit margin. Herbrich and Graepel (2001) is a classical reference.

10) Minor Issues and Missing Citations (R1 & R2 & R3): We thank the reviewers for their comments and suggestions. We have corrected all typos and missing citations in our revisions.

[Meta-Review · NeurIPS 2019]

The paper further develops the direction of proving PAC-Bayesian bounds via techniques to control the Rademacher complexity of the class of distributions with bounded Kullback-Leibler divergence. Connecting these two important tools is of fundamental interest, and the paper has the added benefit of reminding the PAC-Bayesian community of the earlier work in this direction by Kakade, Sridharan and Tewari (2009). For the final version, I would encourage the authors to try to improve the comparisons to related work by carefully considering the remarks of the reviewers. It would also be interesting if they could reflect on whether their results might also be provable using Talagrand's inequality + peeling, as remarked by reviewer #1.